# Fatigue Crack Growth under Non-Proportional Mixed Mode Loading in Rail and Wheel Steel Part 2: Sequential Mode I and Mode III Loading

**Makoto Akama [1],\* and Akira Kiuchi [2]**

[1]    Department of Mechanical Engineering for Transportation, Osaka Sangyo University, Osaka 574-8530, Japan
[2]    Formerly Kobelco Research Institute Inc., Hyogo 651-0073, Japan
\*    Correspondence: akama@tm.osaka-sandai.ac.jp; Tel.: +81-72-875-3001; Fax: +81-72-871-1289

**Abstract:** Rolling contact fatigue cracks in rail and wheel undergo non-proportional mixed mode I/II/III loading. Fatigue tests were performed to determine the coplanar and branch crack growth rates on these materials. Sequential and overlapping mode I and III loading cycles were applied to single cracks in round bar specimens. Experiments in which this is done have been rarely performed. The fracture surface observations and the finite element analysis results suggested that the growth of long (does not branch but grown stably and straight) coplanar cracks was driven mainly by mode III loading. The cracks tended to branch when increasing the material strength and/or the degree of overlap between the mode I and III loading cycles. The equivalent stress intensity factor range that can consider the crack face contact and successfully regressed the crack growth rate data is proposed for the branch crack. Based on the results obtained in this study, the mechanism of long coplanar shear-mode crack growth turned out to be the same regardless of whether the main driving force is in-plane shear or out-of-plane shear.

**Keywords:** non-proportional mixed mode loading; fractography; mode III stress intensity factor; FEA; rail steel; wheel steel

## 1. Introduction

The repeated passage of train wheels over the rails induces rolling contact fatigue (RCF) cracks on both the railhead and the wheel tread. Such surface RCF cracks undergo non-proportional mixed mode I/II/III loading cycles [1–4]. For a period, they grow stably at a shallow angle to the surface, according to what is considered coplanar fatigue crack growth (FCG). Once they have reached a certain length, these cracks start to branch. Coplanar cracks are not a great danger to trains as they flake off, causing the train to have a rougher ride at most. Branch cracks, instead, could lead to catastrophic failure if left to grow. However, the continued growth of coplanar cracks can obscure the branch ones from detection through conventional non-destructive methods. Therefore, the accurate prediction of growth rates for both coplanar and branch cracks is essential to prevent rail and wheel failures and develop cost-effective maintenance strategies.

Many studies have been conducted on the coplanar FCG of rail steel under sequentially applied mode I and II loading cycles. Compared to these mixed mode loading tests, FCG experiments under mixed mode I/III loading are hardly performed. However, several tests under proportional loading or with one mode cyclic and other static have been conducted. Ritchie et al. [5] studied the FCG in mode III AISI 4340 steel in torsional loading, measured on circumferentially-notched specimens and the results were compared with the behavior in mode I. FCG rates in mode III were found to be slower than those in mode I, despite that they were still Paris-type law related to mode III stress intensity

$K_{III}$ range ($\Delta K_{III}$). They proposed a micromechanical model for mode III FCG, in which the crack advance was considered to occur via a mode II coalescence of cracks, initiated at inclusions ahead of the main crack front. Nayeb-Hashemi et al. [6] found no correlation between the FCG rate and the $\Delta K_{III}$ or the plastic strain intensity range ($\Delta \Gamma_{III}$) except when superimposing a static mode I loading to the mode III one on a low-strength AISI 4140 steel. In the latter case, the variability of the FCG rate decreased. Hourlier and Pineau [7] investigated the effects of static mode III on mode I FCG behavior in four materials. They showed that the loading conditions caused two main effects—a considerable reduction in FCG rate and a modification in crack path. They introduced a new criterion based on two main assumptions—fatigue cracking occurred only under the effect of local mode I opening and a fatigue crack grows in a direction where the FCG rate is the maximum. Tarantino et al. [8] applied a novel experimental method to promote mode III coplanar FCG in bearing steel. The method comprised out-of-phase multiaxial fatigue tests, adopting a stress history typical of sub-surface RCF onto specimens containing micronotches. Their analytical model has shown that the typical crack opening values determined by out-of-phase loads can prevent the crack face contact during the RCF loading cycles. Giannella et al. [9] investigated the FCG behavior in cruciform specimens made of Ti6246 by applying a static load along one arm of the specimen and a cyclic load along the other arm. They used an equivalent stress intensity range in the Walker crack growth law that can consider all mode I/II/III loading and determined that a change in the FCG direction occurred depending on the static load levels.

Recently, mixed mode II/III experiments were also performed on a rail steel considering RCF. Bonniot et al. [10] performed the experiments to determine the FCG thresholds and kinetics under loading conditions using asymmetric four-point bending specimens with different angles between the crack front and the shear load. They determined that a coplanar shear-mode FCG occurred with the high loading ranges. The effective stress intensity factors (SIFs) were derived by an inverse method using the relative displacements of the measured crack face. They were found to be 10%–70% lower than the nominal SIF, and a reasonable correlation of the measured FCG rates could be made using the effective SIF.

Akama and Kiuchi [11,12] conducted fatigue tests to determine the coplanar FCG rate on both rail and wheel steel under non-proportional mixed mode I/III loading cycles. They could induce long coplanar cracks under this condition. Moreover, the cracks tended to branch when the degree of overlap ($\delta$) between the mode I and III cycles increased.

As mentioned above, the previous studies about FCG under non-proportional mixed mode loading cycles were apparently focused on the mode I/II loading and coplanar growth in rail steel. Only a detailed study on coplanar and branch FCG under non-proportional mixed mode I/III loading cycles has been performed, by Akama and Kiuchi [11,12], for rail and wheel steel. However, the elucidation of the phenomena inherent to FCG under non-proportional mixed mode I/III loading is not sufficient in their studies. Therefore, in this study, the FCG data obtained from non-proportional mode I/III loading cycles were re-constructed by using novel and reliable equivalent SIF ranges to obtain good correlations. In addition, a finite element analysis (FEA) was performed thoroughly to investigate the crack behavior under these loading cycles.

This paper, which is part 2 of two companion papers, presents FCG under non-proportional mixed mode I/III loading and is organized as follows. Section 2 describes the detailed experimental methods and presents the experimental results. In Section 3, the FEA model for predicting the crack path direction is presented, and the results are provided. Section 4 provides detailed considerations and a discussion by comparing the experimental and FEA results. Finally, the results obtained herein are summarized in Section 5.

## 2. Experiments

### 2.1. Testing Machine and Specimen

The fatigue tests were performed on a servo-hydraulic testing machine, in which capabilities are a maximum tension–compression force of 200 kN, a maximum fully reversed torque of 1000 Nm under load control, and a frequency of 1 Hz in dry conditions.

Circumferentially notched round bar specimens with a 45° starter notch were used. The detailed geometry is depicted in Figure 1. The notch had a 30° groove, a 4.5 mm depth, and a 0.1 mm root radius and was produced at the specimen center by spark erosion. Precracking was not performed, but the growth data within 0.7 mm from the notch tip were discarded.

Four displacement gauges were placed on the knife edges. These edges were attached across the notch on the specimen surface along the circumferential direction at every 90° to measure the compliance, as shown in Figure 2. The specimen was placed into the machine by adjusting the maximum difference from the average value of the strain obtained from these four clip gauges adjusted to within 5%. Prior to the experiments, specimens containing notch depths of 5, 6, 7, and 8.5 mm were prepared, and the relation between load and output of the displacements from the gauges when each specimen was loaded up to 30 kN was investigated. In each case, a linear relation was obtained with no deviation, which was considered to be caused by slip. Additionally, the accuracy was verified using the potential drop method. A personal computer controlled the testing machine by generating the non-proportional loading cycles, as shown in Figure 3, and recorded the data from the strain gauges. The crack length (or depth) (*a*) was calculated via the compliance technique.

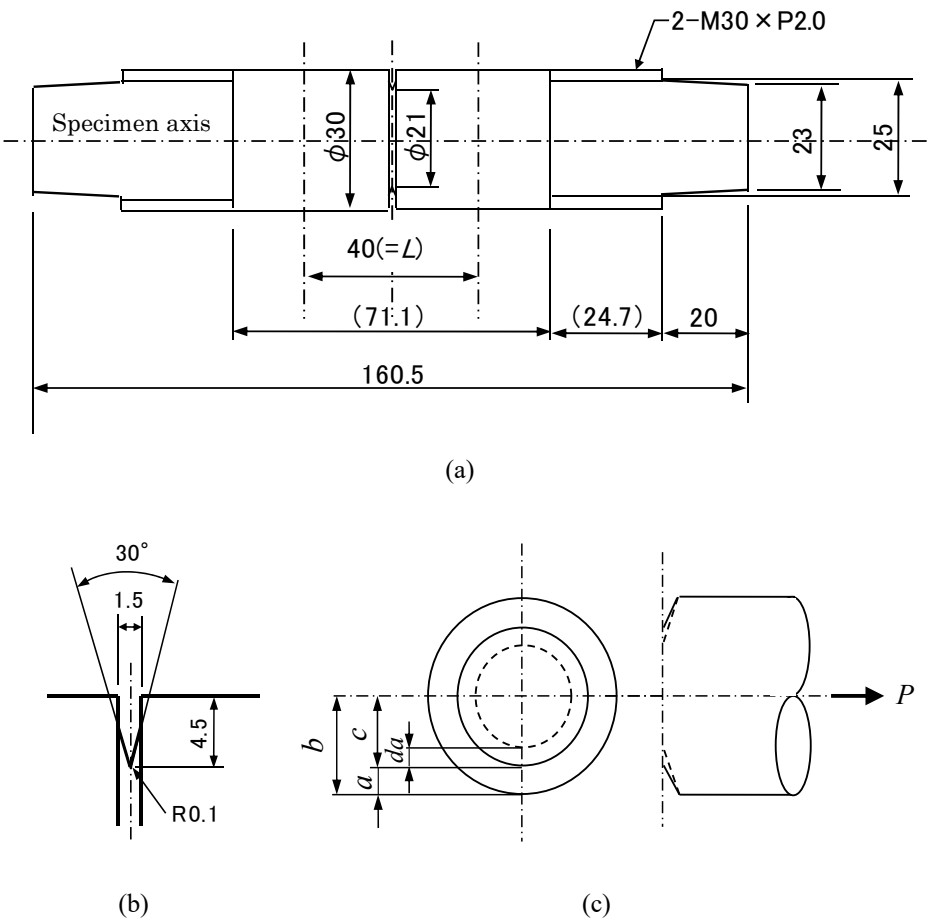

**Figure 1.** (**a**) Whole configuration of the round bar specimen, (**b**) detail of the notch, and (**c**) parameters of the cracked section (all dimensions in mm).

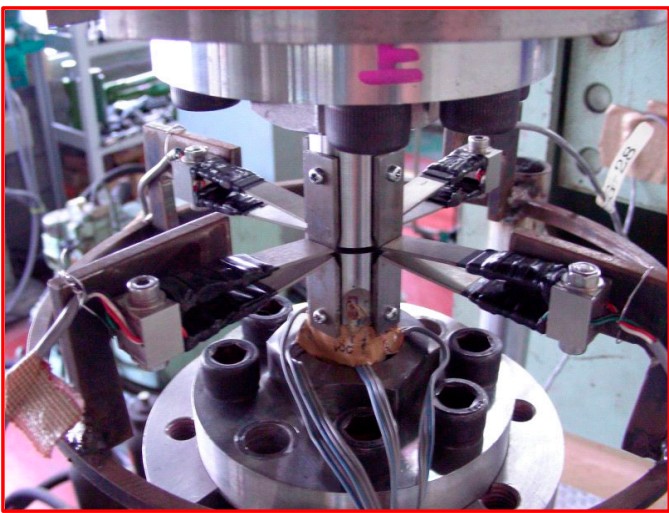

**Figure 2.** Set up of the specimen with displacement gauge attached for the measurement of opening displacements.

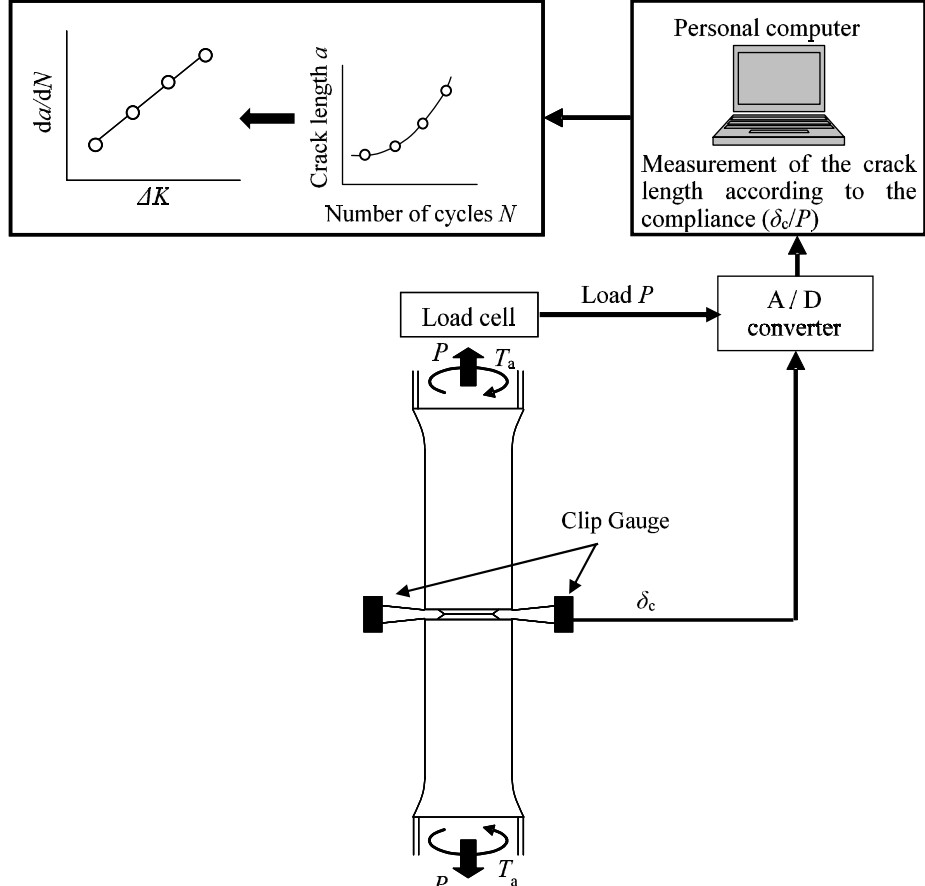

**Figure 3.** Measurement system for the fatigue crack growth rates.

Two rail steels (RP and RF) and one wheel steel (WT) were used as specimen materials. RP and RF were selected to simulate normal and head hardened rails, respectively. Their chemical compositions and mechanical properties are summarized in Tables 1 and 2. The microstructures of RP and WT were normal pearlite, whereas that of RF was fine pearlite with an average lamellar spacing of about 100 nm. The specimens were directly collected from real rails and wheels.

**Table 1.** Chemical composition (wt%).

| Material | C | Si | Mn | P | S |
|---|---|---|---|---|---|
| Rail steel, RP | 0.68 | 0.26 | 0.93 | 0.016 | 0.01 |
| Rail steel, RF | 0.79 | 0.17 | 0.82 | 0.019 | 0.01 |
| Wheel steel, WT | 0.65 | 0.26 | 0.73 | 0.016 | 0.01 |

**Table 2.** Mechanical properties.

| Material | Ultimate Tensile Strength (MPa) | 0.2% Proof Stress (MPa) |
|---|---|---|
| Rail steel, RP | 934 | 511 |
| Rail steel, RF | 1214 | 802 |
| Wheel steel, WT | from 981 to 1030 | from 618 to 657 |

*2.2. Loading History*

The loading history simulated the one experienced by RCF cracks in the presence of a fluid, as obtained by FEA [1,3] and represented in Figure 4, which also shows $\delta$ between $K_I$ and $K_{III}$.

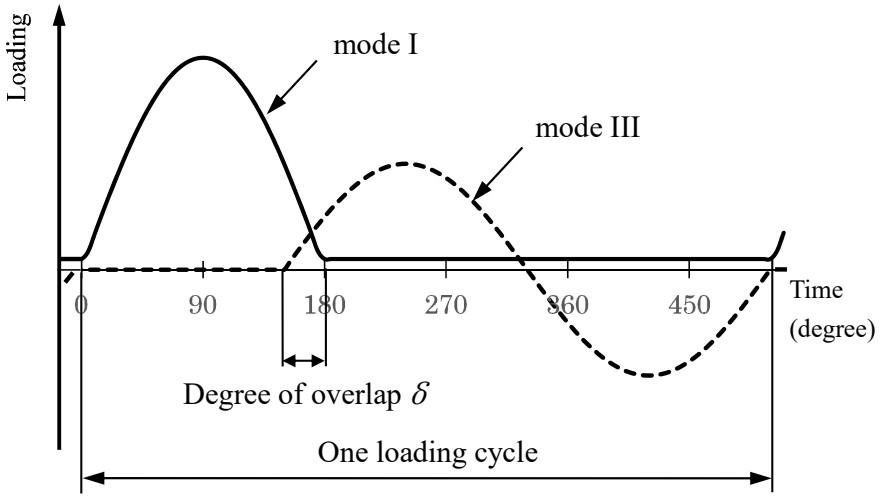

**Figure 4.** A loading cycle applied under the condition of $\Delta K_{III}/\Delta K_I = 1.0$ and $\delta = 30°$.

*2.3. Calculation of the Stress Intensity Factors*

$K_I$ and $K_{III}$ were calculated as follows [13]:

$$K_I = f(x)\sigma \sqrt{\pi a} \tag{1}$$

and

$$K_{III} = g(x)\tau \sqrt{\pi a} \tag{2}$$

where $\sigma$ and $\tau$ are the normal and out-of-plane shear stress, respectively, applied to the crack. The functions $f(x)$ and $g(x)$ are expressed as follows:

$$f(x) = \frac{1}{2} \sqrt{x}\left(1 + \frac{1}{2}x + \frac{3}{8}x^2 - 0.363x^3 + 0.731x^4\right) \tag{3}$$

and

$$g(x) = \frac{3}{8} \sqrt{x}\left(1 + \frac{1}{2}x + \frac{3}{8}x^2 + \frac{5}{16}x^3 + \frac{35}{128}x^4 + 0.208x^5\right) \tag{4}$$

where *x* indicates the *c/b* represented in Figure 1c, *b* is the radius of the specimen, and *c* is the radius of the remaining ligament.

## 2.4. Experimental Conditions

The experiments were conducted on five RP—three RF and one WT specimen—hereafter referred to as RP1, RP2, RP3, RP4, and RP5; RF1, RF2, and RF3; and WT1. Table 3 reports the $\Delta K_{III}/\Delta K_I$ ratios and $\delta$ values for each experiment. In all tests, the crack faces were slightly pulled apart so that the stress ratios of mode I ($R_I$) and III ($R_{III}$) loading were 0.05 and −1, respectively. RP1, RP2, RP3, RF1, RF2, and WT1 were designed to provide mainly coplanar FCG rate data, whereas RP4, RP5, and RF3 were conducted to obtain branch FCG rate data. RP1 was carried with $\delta = 90°$ at the first step. Once the crack length reached about 7 mm, it was interrupted and continued by changing $\delta$ to 30°. In a similar way, RP3, RF1, RF2, and WT1 were performed by changing $\delta$. After the tests, the specimens were broken by applying a large tensile force to observe the fracture surfaces.

**Table 3.** Testing conditions for mixed mode I/III loading tests.

| Exp. No. | $\Delta K_{III}/\Delta K_I$ | $\Delta$ (degree) |
|---|---|---|
| RP1 | 1.0 | $90 \rightarrow 30$ |
| RP2 | 1.5 | 90 |
| RP3 | 1.5 | $30 \rightarrow 120$ |
| RP4 | 1.0 | 180 |
| RP5 | 1.5 | 180 |
| RF1 | 1.0 | $90 \rightarrow 30$ |
| RF2 | 1.5 | $60 \rightarrow 30$ |
| RF3 | 1.0 | 180 |
| WT1 | 1.0 | $90 \rightarrow 60$ |

## 2.5. Experimental Results

Figure 5 schematically represents the main coplanar crack plane and the angles between this plane and the branch crack plane when branching occurred. In this figure, $\theta$ and $\varphi$ are the twisting and deflection angles, respectively.

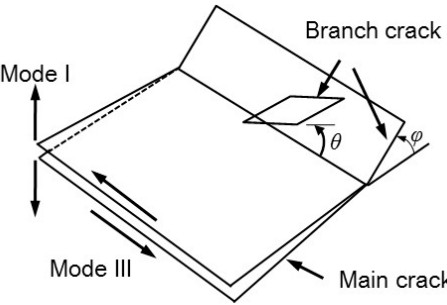

**Figure 5.** Branch crack at a main crack and branch angle definition.

The coplanar cracks grew almost straight (i.e., $\theta = \varphi = 0°$) in RP1 and RP2 but branched in RP3 at about $\theta = 45°$ and $\varphi = 0°$, when $\delta$ was 120°. In RP4 and RP5, the branching occurred and resulted in factory-roof fractures (i.e., associated with ridges and valleys). In RF1 and RF2, little factory-roof fractures were observed on the crack faces when $\delta$ was 90° and 60°, respectively, but coplanar cracks grew in both experiments when it was reduced to 30°. The crack branched in RF3. In WT1, the fracture surface was flat, the crack grew coplanarly, and no branching was observed. Figure 6 shows the macroscopic fracture surfaces obtained in RP2, RP5, RF2, and WT1.

### 2.5.1. Coplanar Crack Growth Rate

When mode III and mode I are mixed, the equivalent SIF range at $\varphi = 0°$ in Figure 5 can be expressed as [14]

$$\Delta K_s = 0.5\left\{\Delta K_I^2(1 - 2v)^2 + 4\Delta K_{III}^2\right\}^{0.5} \tag{5}$$

where $v$ is the Poisson ratio. When $v = 0.3$, it becomes

$$\Delta K_s = 0.5\left\{0.16\Delta K_I^2 + 4\Delta K_{III}^2\right\}^{0.5} \tag{6}$$

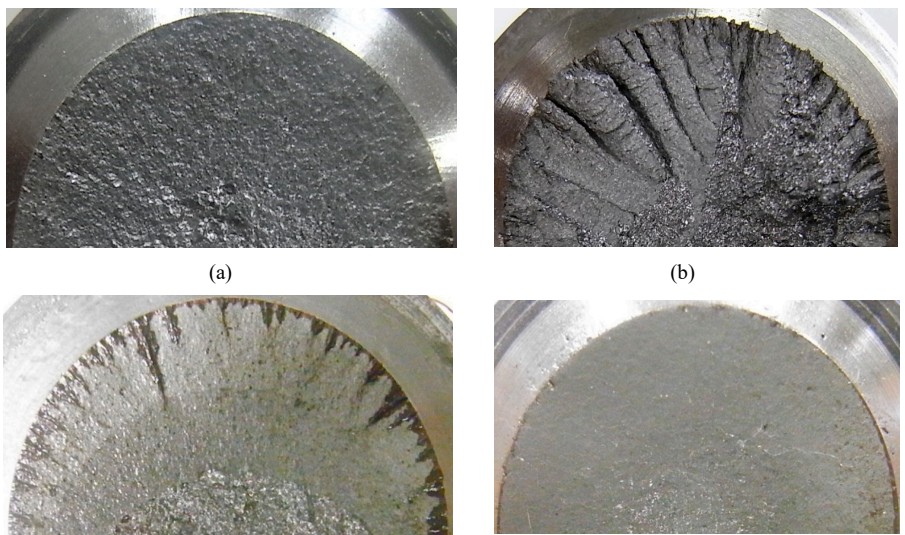

**Figure 6.** Macroscopic fracture surface of (**a**) RP2 ($\Delta K_{III}/\Delta K_I = 1.5$, $\delta = 90°$), (**b**) RP5 ($\Delta K_{III}/\Delta K_I = 1.5$, $\delta = 180°$), (**c**) RF2 ($\Delta K_{III}/\Delta K_I = 1.5$, $\delta = 60°\rightarrow30°$) and (**d**) WT1 ($\Delta K_{III}/\Delta K_I = 1.0$, $\delta = 90°\rightarrow60°$).

Figure 7 shows the growth rate data obtained for RP, consisting of the RP1, RP2, and RP3 results, with respect to $\Delta K_s$. The data are observed to be divided into two groups by $\delta$. When the Paris-type law was applied to the data for the same $\delta$, the following equations were obtained:

$$\frac{da}{dN} = C(\Delta K_s)^m \tag{7}$$

where $N$ is the number of cycles. For $\delta = 90°$, $C = 2.52 \times 10^{-11}$ and $m = 2.93$ and for $\delta = 30°$, $C = 1.15 \times 10^{-10}$ and $m = 2.26$. The $R$-squared value, i.e., the coefficient of determination ($R^2$), is 0.973 and 0.917. In each case, a good correlation could be obtained even when the $\Delta K_{III}/\Delta K_I$ differed.

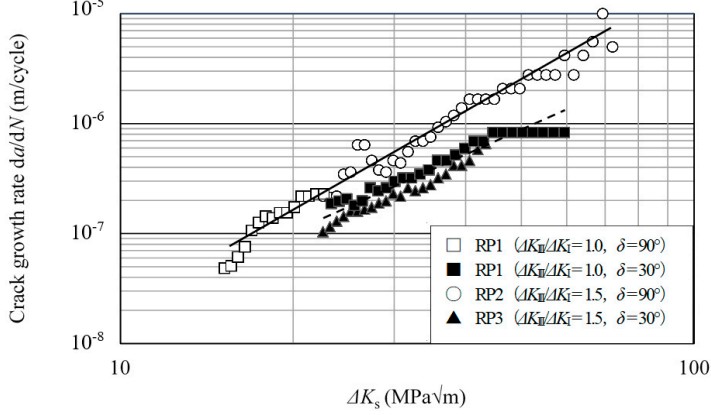

**Figure 7.** Coplanar crack growth rates against $\Delta K_s$ for RP.

Next, the growth rate data obtained for RF, comprising of the RF1 and RF2 results, were correlated with respect to $\Delta K_s$, as shown in Figure 8. In this case, the correlation was poor. The growth law was as follows:

$$\mathrm{d}a/\mathrm{d}N = 7.40 \times 10^{-10}(\Delta K_s)^{1.62} \ (R^2 = 0.638) \tag{8}$$

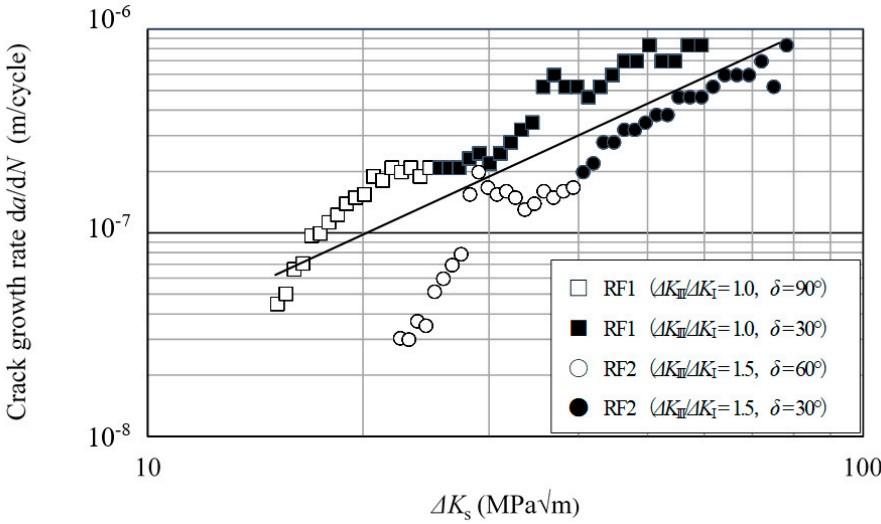

**Figure 8.** Coplanar crack growth rates against $\Delta K_s$ for RF.

Although not shown in the figure, when $\delta$ was constant, the growth rates increased with $\Delta K_{III}/\Delta K_I$. When $\Delta K_{III}/\Delta K_I$ was constant, the growth became faster owing to the increased $\delta$.

Figure 9 shows the FCG rate data as a function of $\Delta K_s$ when the $\Delta K_{III}/\Delta K_I$ was 1.0, for all the specimens. WT exhibited the highest growth rate, followed by RP and RF.

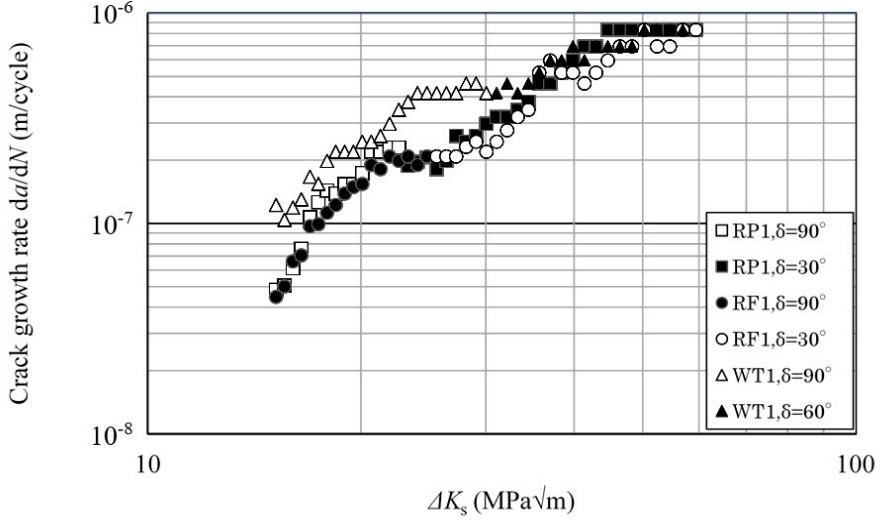

**Figure 9.** Comparison of crack growth rates against $\Delta K_s$ for RP, RF, and WT ($\Delta K_{III}/\Delta K_I = 1.0$).

### 2.5.2. Branch Crack Growth Rate

In the mixed mode I/III loading, the cracks grown may branch to form torsional facets. In such a case, the cracks form perpendicularly to the maximum principal stress and the equivalent SIF range has been proposed as follows [15]:

$$\Delta K_t = 0.8\Delta K_I + 0.5\{0.16\Delta K_I^2 + 4\Delta K_{III}^2\}^{0.5} \tag{9}$$

under the condition of $\nu = 0.3$. The combined growth data obtained from RP4 and RP5 as a function of $\Delta K_t$ are plotted in Figure 10. When $\Delta K_{III}/\Delta K_I$ was large (i.e., in the RP5 case), the growth rates went down. This happened because Equation (9) does not consider the attenuation of $\Delta K_{III}$ due to the crack face contact. This attenuation was considered in the previously performed mixed mode I/III test with $R_I = R_{III} = 0.05$ and the equivalent SIF range ($\Delta K_{eq}$) that was weighted to $\Delta K_{III}$, as in the following equation [16], was proposed:

$$\Delta K_{eq} = \left\{ \Delta K_I^2 + 0.2 \Delta K_{III}^2 \right\}^{0.5} \tag{10}$$

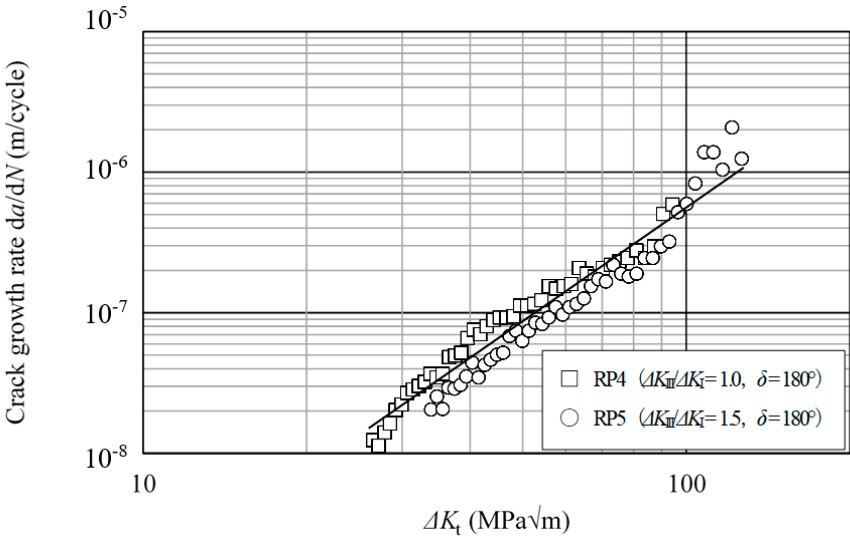

**Figure 10.** Branch crack growth rates against $\Delta K_t$ for RP.

When the growth rates were plotted against $\Delta K_{eq}$, a good correlation was obtained even when $\Delta K_{III}/\Delta K_I$ and $\delta$ were extensively changed. Therefore, also in this study, the growth rates for RP and RF were plotted against $\Delta K_{eq}$, and the results for RP are shown in Figure 11, revealing a fairly good correlation. The growth rate in RP was expressed via the Paris-type law as follows:

$$da/dN = 1.84 \times 10^{-12} (\Delta K_{eq})^{3.29} \quad (R^2 = 0.960) \tag{11}$$

Although not shown, the growth law for RF was

$$da/dN = 4.66 \times 10^{-13} (\Delta K_{eq})^{3.71} \quad (R^2 = 0.987) \tag{12}$$

As mentioned previously, coplanar growth was considered to occur in RF1 and RF2 when $\delta$ was 30°, and the growth rates were plotted with $\Delta K_s$. However, the correlation was poor in these cases. This was probably due to the effect of small factory-roof fractures that formed on the fracture surfaces under the initial conditions. Although the coplanar growth occurred when $\delta$ was 30° for both specimens, an effect of the contact at the factory-roof fracture surfaces on the growth rates was observed throughout the experiments. Therefore, all data were correlated with $\Delta K_{eq}$, and the results are shown in Figure 12. A better correlation was obtained compared with the case of plotting with $\Delta K_s$. The growth law was as follows:

$$da/dN = 4.51 \times 10^{-11} (\Delta K_{eq})^{2.60} \quad (R^2 = 0.823) \tag{13}$$

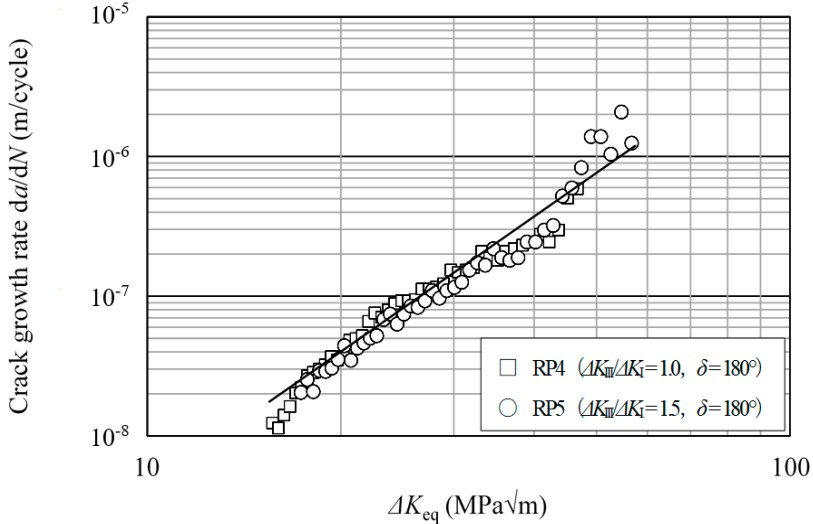

**Figure 11.** Branch crack growth rates against $\Delta K_{eq}$ for RP.

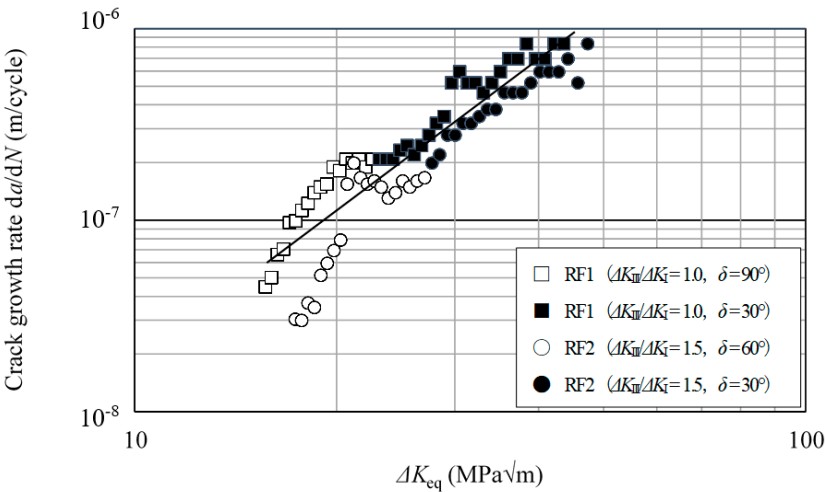

**Figure 12.** Branch crack growth rate against $\Delta K_{eq}$ for RF.

### 2.5.3. Fractography

After the experiments, scanning electric microscope (SEM) observations were performed to unravel the FCG mechanism from the fracture surfaces of the specimens. The SEM images of the fracture surfaces of the RP1, RP2, and RP3 specimens, at various growth times, are shown in Figures 13–15. Fujii et al. [17] performed fatigue tests under mode II loading, with high growth rates and short crack lengths, and observed severe rub marks and many wear particles and oxides on the fracture surfaces. In RP1 and RP2, however, there was no evidence of contact between the crack surfaces at all growth periods, and no oxide debris generated from the surface appeared. At high magnifications, microcleavage and rippling could be observed, implying that the attenuation due to surface friction was very limited. As in previous non-proportional mixed mode I/II loading experiments [18,19], clear striation patterns were seldom seen. Because there was no contact between the crack faces, the striation patterns were not worn out by the mating face.

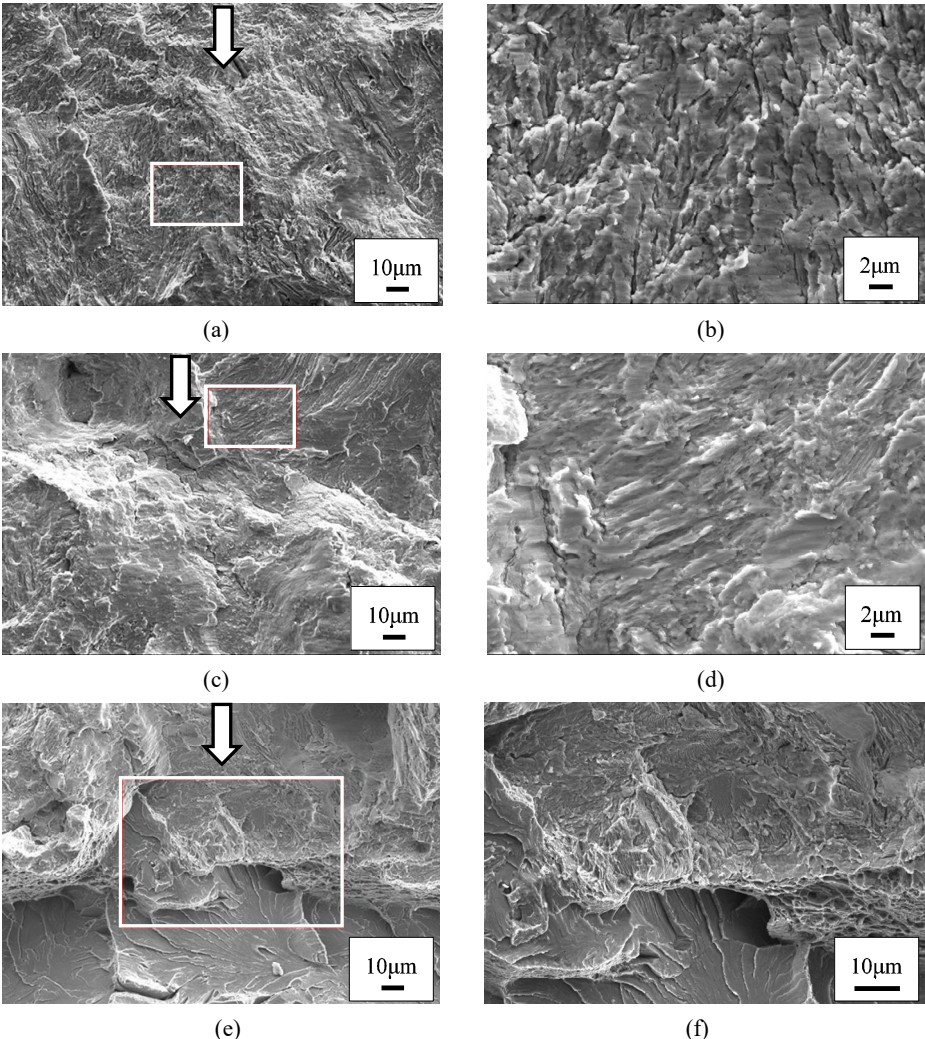

**Figure 13.** Fracture surface of RP1 observed via scanning electric microscope (SEM) (arrows indicate the direction of the crack growth). (**a**) 2 mm from the notch tip ($\delta = 90°$), (**b**) an enlarged view indicated by the frame in (**a**), (**c**) 4.8 mm from the notch tip ($\delta = 30°$), (**d**) frame in (**c**), (**e**) 10.5 mm from the notch tip ($\delta = 30°$), and (**f**) frame in (**e**).

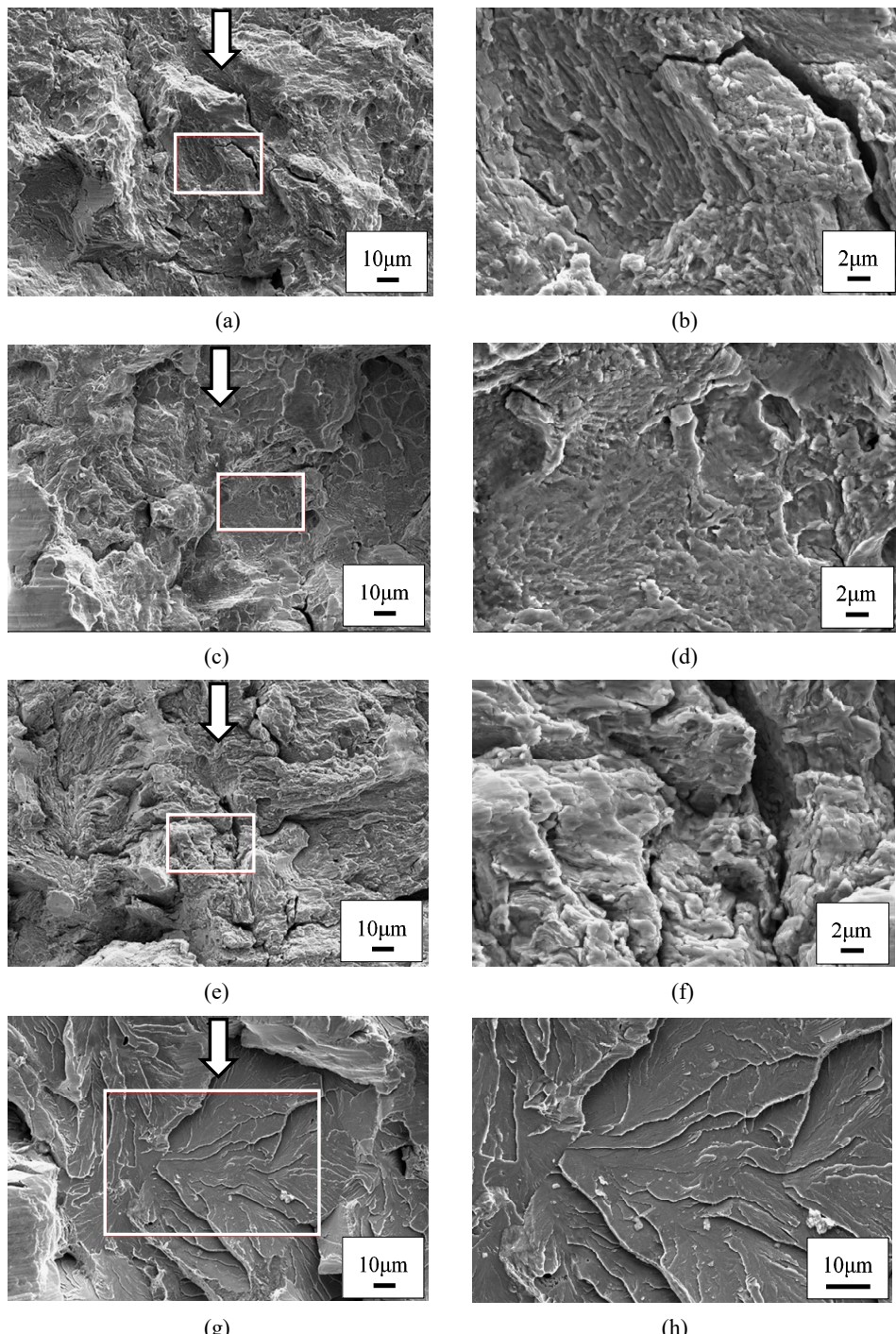

**Figure 14.** Fracture surface of RP2 observed via SEM (arrows indicate the direction of the crack growth). (**a**) 2 mm from the notch tip, (**b**) an enlarged view indicated by the frame in (**a**), (**c**) 4 mm from the notch tip, (**d**) frame in (**c**), (**e**) 6.8 mm from the notch tip, (**f**) frame in (**e**), (**g**) 8 mm from the notch tip, and (**h**) frame in (**g**).

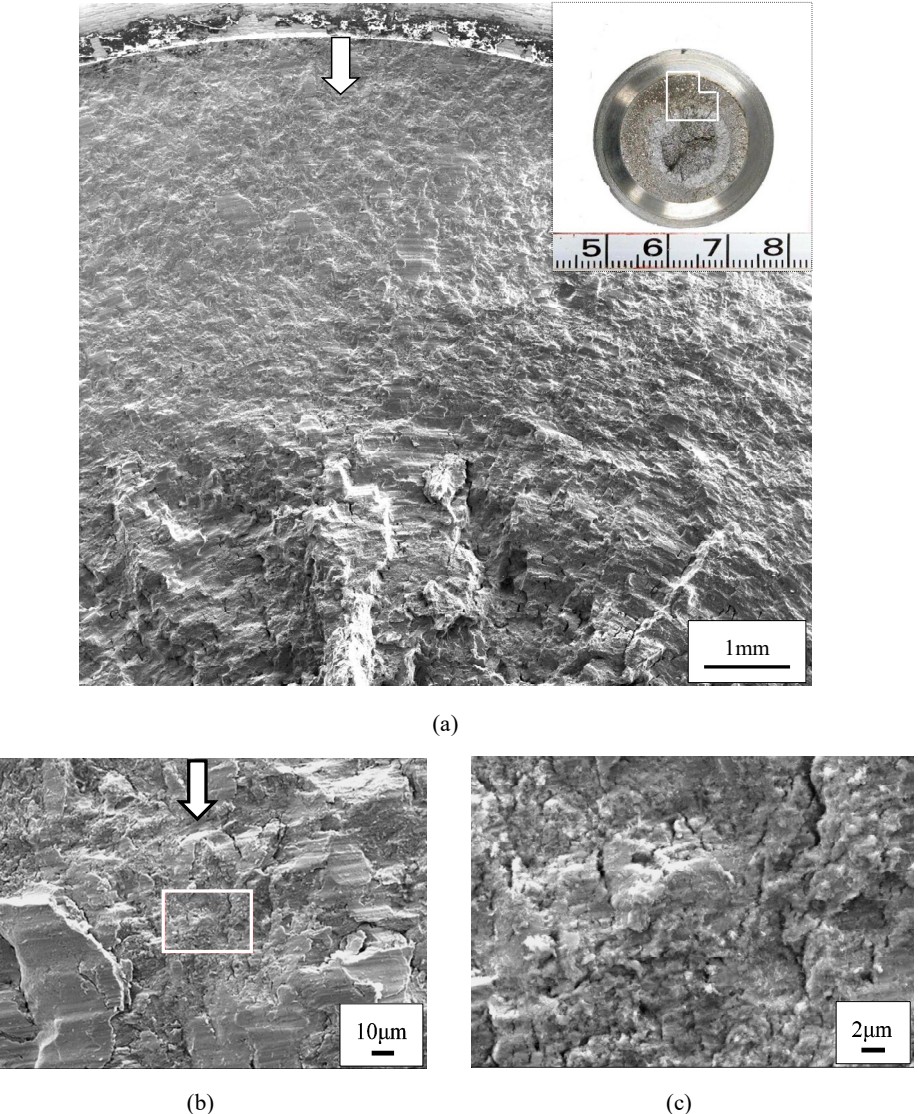

**Figure 15.** Fracture surface of RP3 observed using a SEM (arrows indicate the direction of the crack growth). (**a**) Macroscopic appearance of the fracture surface, (**b**) 3 mm from the notch tip ($\delta = 30°$), and (**c**) an enlarged view indicated by a frame in (**b**).

## 3. Finite Element Analysis

### 3.1. Procedure

A series of elasto-plastic FEA was performed using the commercial FEA code MARC to elucidate the FCG characteristics observed in the mixed mode I/III loading tests and predict the FCG direction, which is $\theta$ and $\varphi$ in Figure 5. Therefore, a stationary crack was considered. The three-dimensional (3D) finite element mesh of the round bar specimen and the boundary conditions applied are shown in Figure 16. The plasticity was considered highly localized near the notch under the loading, while other specimen regions remained elastic. Therefore, the area close to the symmetry section for the axial direction was enough for the mesh. The mesh density increased at the crack tip where, for the radial direction, the size of an 8-node brick element was 10 μm, which was considered well included in the plastic zone ahead. The total numbers of elements and nodes were 14,001 and 15,522, respectively. This mesh and the loading procedure confirmed that the nominal stress value in the ligament ahead of the crack was accurate.

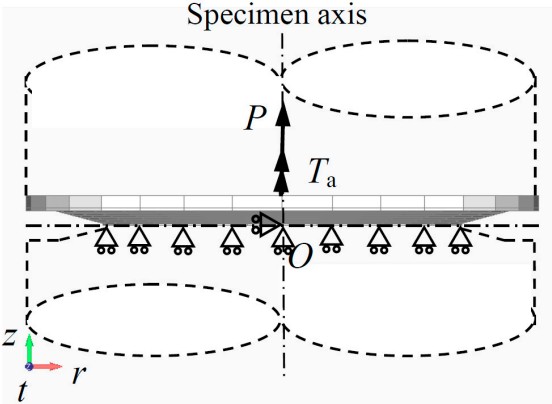

**Figure 16.** Mesh used for the analyses, and applied loads and boundary conditions.

At the center of the upper surface of the mesh, a pilot node was connected to the remaining nodes of that surface through a multipoint constraint; force ($P$) and torque ($T_a$) were applied to the node; their ranges $\Delta P$ and $\Delta T_a$ were equal to 55 kN ($R_I = 0.05$) and 525 Nm ($R_{III} = -1$), respectively. If $\Delta P$ and $\Delta T_a$ were equal to 55 kN and 525 Nm, respectively, $\Delta K_{III}/\Delta K_I$ became 1.5. The fixed boundary conditions applied on the lower surface of the mesh were of the same as the symmetry condition for the axial direction. The $a$ value was set at 4.6 mm for the cases of $\delta = 30°$, $90°$, and $180°$ and at 7 mm for $\delta = 120°$, representing the actual RP3 test.

In this study, the model combining nonlinear kinematic hardening rule with the isotropic hardening rule developed by Chaboche and Lemaitre [20] (C & L model) was employed.

$$^{ti+\Delta ti}\sigma_y = {}^0\sigma_y + Q\left\{1 - \exp\left(-B^{ti+\Delta ti}\bar{e}^p\right)\right\} \tag{14}$$

and

$$\mathrm{d}\alpha = \frac{2}{3}h\mathrm{d}e^p - \zeta\alpha\mathrm{d}\bar{e}^p \tag{15}$$

where $^{ti+\Delta ti}\sigma_y$ is the updated yield stress at time $t_i + \Delta t_i$, $^0\sigma_y$ is the initial yield stress, $Q$, $B$, $h$, and $\zeta$ are material constants, $^{ti+\Delta ti}\bar{e}^p$ is the accumulated effective plastic strain at $t_i + \Delta t_i$, $\alpha$ is the shift of the yield surface center, $e^p$ is the plastic strain, and d implies increment. The material constants for RP and RF are summarized in Table 4 along with Young's modulus and Poisson ratio. The FEA was performed on these two rail steels to clarify the material effect on the FCG rate.

**Table 4.** Material properties used in finite element analysis (FEA).

| Material | $E$ (MPa) | $\nu$ | $^0\sigma_y$ (MPa) | $Q$ | $b$ | $h$ (MPa) | $\zeta$ |
|----------|-----------|-------|--------------------|-----|-----|-----------|---------|
| RP | 183,008 | 0.3 | 508 | −208 | 24.2 | 85,248 | 193 |
| RF | 182,778 | 0.3 | 684 | −264 | 1.27 | 88,615 | 185 |

### 3.2. Analytical Results

The contour plots of total displacement and *tz*-stress on the shapes deformed under the positions of the loading cycle for the RP case are shown in Figure 17. The *rtz* represents global cylindrical coordinate system whose origin coincides with specimen center *O*. The following evaluations were performed during the 160th loading cycle because the stress states have converged at that cycle.

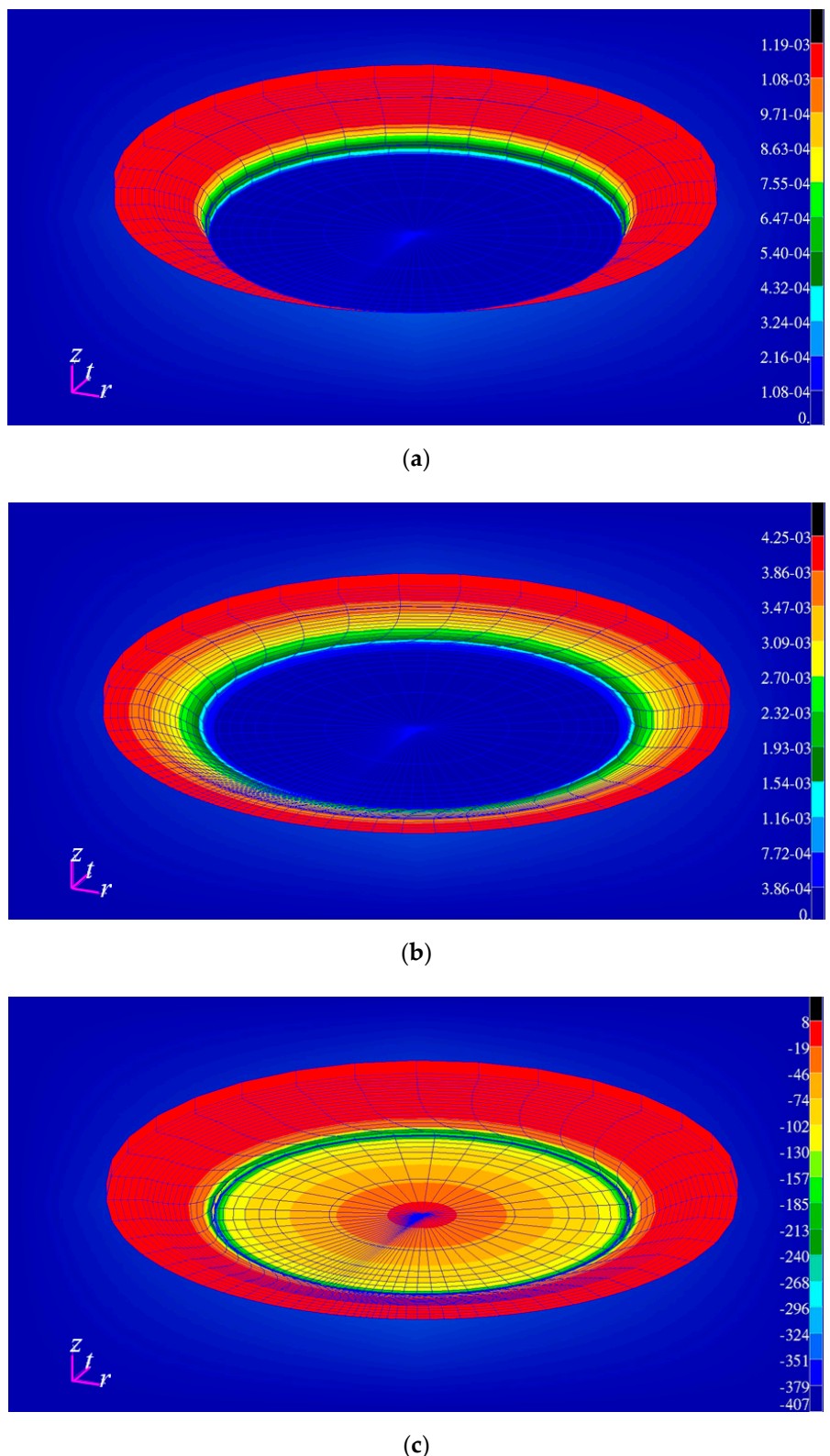

**Figure 17.** The contour plots of total displacement and *tz*-stress on the shapes deformed under the positions of the loading cycle for the RP case ($\Delta K_{III}/\Delta K_{I}$ = 1.5, $\delta$ = 90°), (**a**) maximum deformation by mode I loading, (**b**) and (**c**) maximum deformations by mode III loading. (**a**) and (**b**) depict total displacements and (**c**) depicts *tz*-stress. Notably, the contour levels for displacement in mm and for stress in MPa. Deformation amplifications are 1000×.

### 3.2.1. Planes of Maximum Normal and Shear Stress Ranges

First, the maximum tangential stress range was investigated to elucidate the effect of $\delta$ on crack branching direction [11]. The analysis was suggested to be based on an elasto-plastic stress field [21]. The normal stress range ($\Delta\sigma$) and the shear stress range ($\Delta\tau$) level on every plane were investigated. Such a plane was expressed by $\theta$, $\varphi$, and $\psi$ that are the rotation angles around the axes of the local cylindrical coordinate system $R$, $T$, and $Z$ whose origin coincides with the center of the crack tip element (see Figure 18). Underlying assumption is that if the cracks grow by a tensile mode, the growth direction should be determined by the $\Delta\sigma$, whereas when the cracks grow by a shear mode, the direction is determined by the $\Delta\tau$ near the crack tip. Considering that $\Delta\tau_{ZR}$ was very small under the applied loading cycles, $\Delta\tau$ was represented by a range of $\Delta\tau_{ZT}$ values.

The variations of $\Delta\sigma$ and $\Delta\tau$ due to $\theta$ for the different values of $\delta$ for the RP case were indicated on the $\varphi = \psi = 0$ plane in Figure 19. $\Delta\sigma$ and $\Delta\tau$ at each $\theta$ were divided by the maximum of $\Delta\tau$ ($\Delta\tau_{max}$) because the absolute values depend on the distance from the crack tip and are not important. When $\delta$ increases, the relative $\Delta\sigma$ value also increases and the maximum of $\Delta\sigma$ ($\Delta\sigma_{max}$) plane turns toward the branch direction, whereas the $\Delta\tau_{max}$ plane remains on the coplanar plane. In the calculation results, both $\Delta\sigma_{max}$ and $\Delta\tau_{max}$ planes were slightly oriented toward the $\varphi$ direction within 7° for all cases.

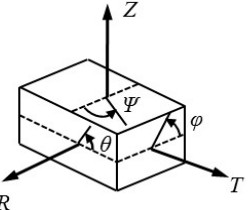

**Figure 18.** Definition of local cylindrical coordinate system *RTZ*. Notably, $\theta$ and $\varphi$ coincide with the branch angles defined earlier in Figure 5.

### 3.2.2. Planes of Maximum Principal Stress

The criterion for multiaxial FCG proposed by Schöllmann et al. [22] is based on the maximum principal stress (MPS)—the crack will grow radially from the crack front in the direction that is perpendicular to the MPS, $\sigma_1$, on a virtual cylindrical surface around the crack front (see Figure 5). Therefore, herein, the maximum values of $\sigma_1$ ($\sigma_{1\,max}$) and directions perpendicular to $\sigma_{1\,max}$ during one loading cycle were investigated for each loading condition.

The $\sigma_{1\,max}$, middle ($\sigma_2$), and minimum ($\sigma_3$) principal stresses for the different values of $\delta$ for the RP case are shown in Figure 20, and in Table 5, the planes of $\sigma_{1\,max}$ are indicated for this case. As shown, as $\delta$ increases, the $\sigma_{1\,max}$ plane turns toward the branch direction.

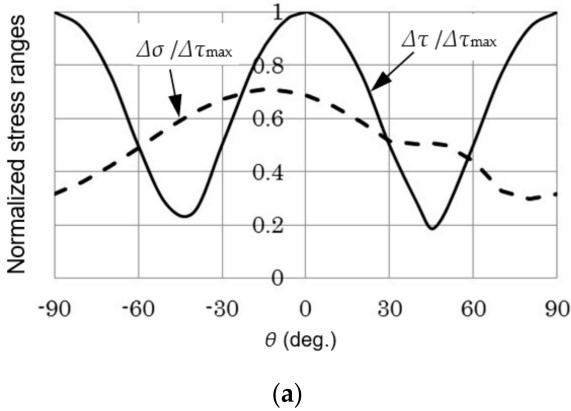

(a)

**Figure 19.** *Cont.*

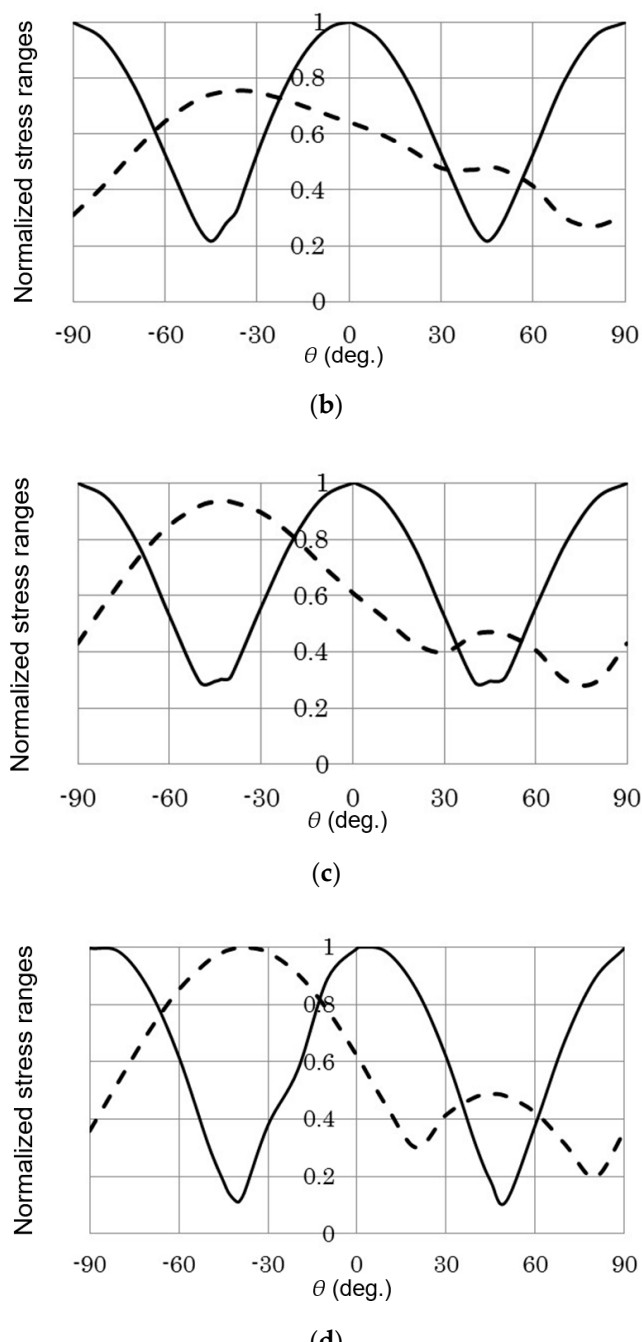

**Figure 19.** Shear stress range and normal stress range at each angle $\theta$ for the case of RP steel ($\Delta K_{III}/\Delta K_I = 1.5$, $R_I = 0.05$, $R_{III} = -1$). (**a**) $\delta = 30°$, (**b**) $\delta = 90°$, (**c**) $\delta = 120°$, and (**d**) $\delta = 180°$.

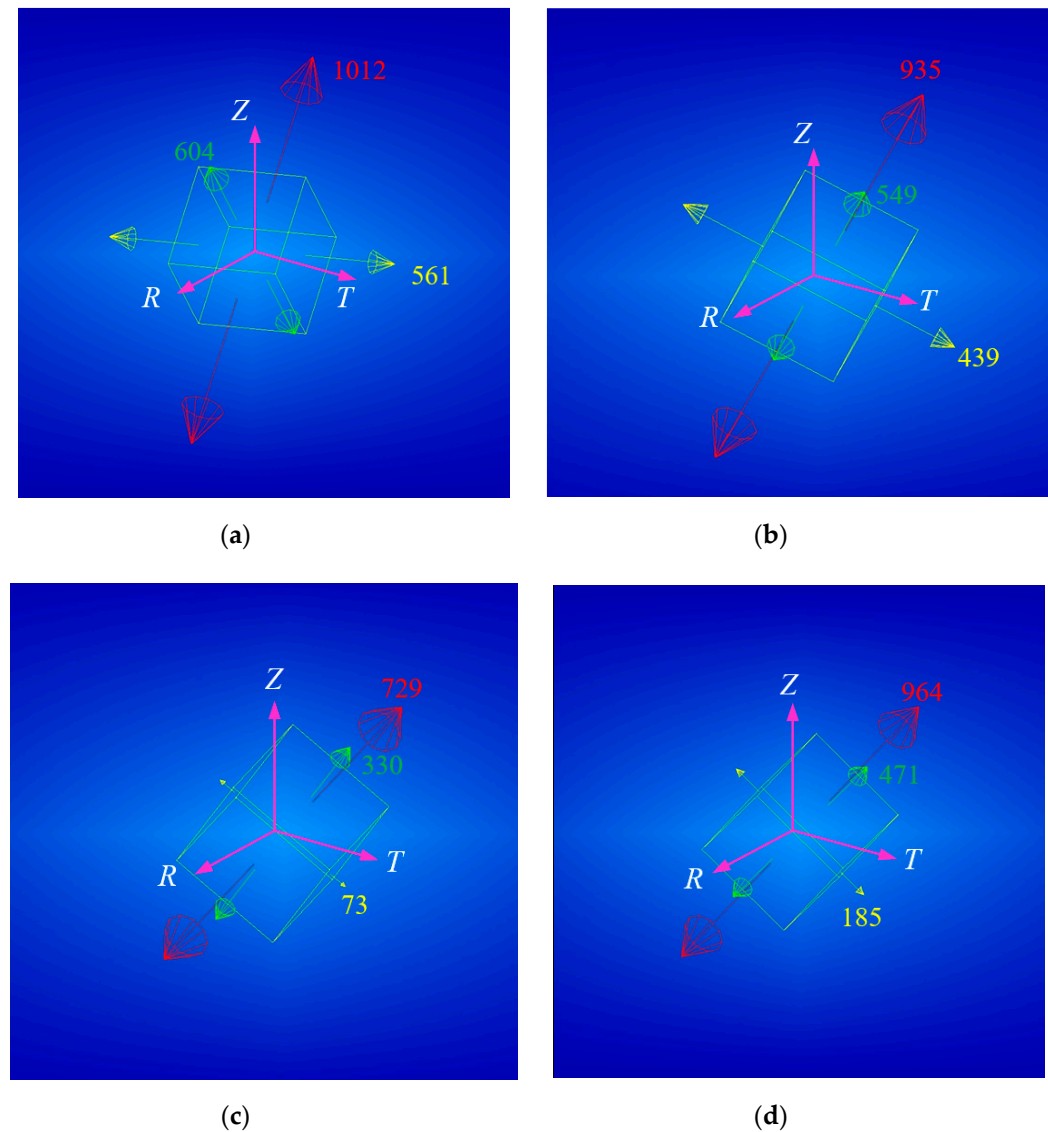

**Figure 20.** Display of principal stress tensors when $\sigma_1$ was the maximum for the case of RP steel ($\Delta K_{III}/\Delta K_I = 1.5$, $R_I = 0.05$, $R_{III} = -1$). (**a**) $\delta = 30°$, (**b**) $\delta = 90°$, (**c**) $\delta = 120°$, and (**d**) $\delta = 180°$. Red arrow indicates $\sigma_{1\,max}$, and green and yellow arrows indicate $\sigma_2$ and $\sigma_3$, respectively. Their lengths represent the relative magnitudes and the figures are in MPa.

**Table 5.** Planes of $\sigma_{1\,max}$ for the case of RP steel ($\Delta K_{III}/\Delta K_I = 1.5$, $R_I = 0.05$, $R_{III} = -1$).

| $\delta$(deg.) | 30 | 90 | 120 | 180 |
|---|---|---|---|---|
| $\theta$(deg.) | −17 | −27 | −42 | −41 |
| $\varphi$(deg.) | −4 | −3 | 9 | 4 |

### 3.2.3. Crack Tip Opening Displacement

The FEA results were used to clarify the crack tip opening displacement (CTOD) of RP and RF during the 160th loading. Figure 21 shows the variations of CTOD/2 at several positions from the crack tip ($D$) for both steels. The values are smaller for RF; in particular, when the tensile load decreased, the time in degrees was greater than 180°, half of the CTODs became smaller than $4 \times 10^{-5}$ mm in a wide range (50 µm $\leq D \leq$ 100 µm).

## 4. Discussion

In the case of RP and WT, a coplanar growth was obtained when $\delta$ was smaller than or equal to 90° regardless of the $\Delta K_{III}/\Delta K_I$ value and the growth rate was well correlated with $\Delta K_s$, as shown in Figure 9. According to Equation (6), $\Delta K_I$ was equal to only 4% of $\Delta K_{III}$, suggesting a strong dependence of the FCG rate on the mode III loading. Conversely, in the RF case, some factory-roof fractures were observed at $\delta = 60°$ and 90°, and no good correlation with $\Delta K_s$ was obtained.

The FEA results shown in Figure 21 indicated that the CTOD variation in one loading cycle was smaller for RF than for RP under the same loading condition. In non-proportional mixed mode loading cycles, fatigue cracks follow the direction of the $\Delta\sigma_{max}$ or $\Delta\tau_{max}$ plane depending on if these growth rates on these planes is faster [23]. When the CTOD is small, the actual crack faces have irregularities and, hence, are likely to make contact. When the crack faces make contact, the friction can make the coplanar FCG rate smaller relative to the branch FCG rate. Therefore, factory-roof fractures, which are traces of branch FCG, should have appeared.

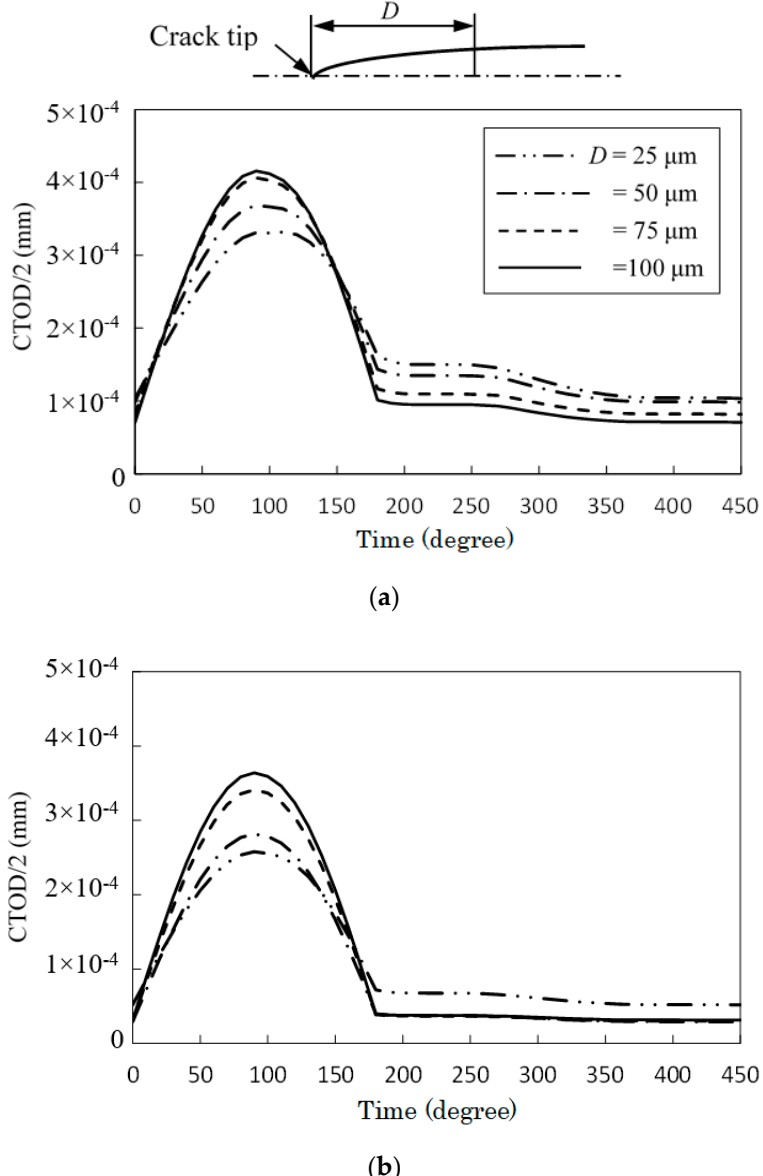

(a)

(b)

**Figure 21.** Variations of half the crack tip opening displacements behind the crack tip at 160th cycle of (a) RP and (b) RF. ($\Delta K_{III}/\Delta K_I = 1.5$, $\delta = 90°$).

When $\delta$ increased to 120°, branch cracks also grew even in RP. According to the FEA results shown in Figure 19, increasing $\delta$ increased $\Delta\sigma_{max}$ with respect to $\Delta\tau_{max}$ and turned it toward the branch direction. Conversely, $\Delta\tau_{max}$ remained on the coplanar direction. The coplanar FCG rates under the loading cycles adopted herein are considered to have a strong dependence on $K_{III}$. Therefore, it should also have a relatively strong dependence on $\Delta\tau_{max}$, while the branch FCG rates are generally considered to have a relation with $\Delta\sigma_{max}$. These facts conclude that cracks tend to branch when $\delta$ increases.

We determined that when $\delta$ increases, the $\sigma_{1\,max}$ plane turns toward the branch direction, as indicated in Table 5. However, it may be unsuitable to use the MPS criterion to predict the FCG direction under the loading conditions adopted herein. This criterion assumes that the crack grows in mode I. When the coplanar FCG planes were observed using SEM, no clear striation, which is evidence of FCG caused by mode I loading, was observed.

The SEM observations provided no evidence to support crack face contact in case of RP at $\Delta K_{III}/\Delta K_I = 1.5$ and $\delta = 90°$. This was confirmed by the FEA results, in which the crack was always widely open (see Figure 21a). Rubbing has been suggested as the reason for the FCG termination. Under the testing conditions in the present study, the crack faces were opened during the experiments. If the crack face contacts with its mating face and the generated friction attenuates the mode III loading, the crack is considered to be arrested. If there is no contact, a long shear mode crack becomes possible under this loading condition.

Although the branch FCG rates were plotted against $\Delta K_t$, a good correlation was not obtained because $\Delta K_t$ does not consider the $\Delta K_{III}$ attenuation due to the crack face contact. Therefore, $\Delta K_{eq}$, which includes this attenuation, was proposed, providing a FCG law with a fairly good correlation. Even for the RF steel, where some factory-roof fractures arose on the fracture surface, a better correlation was obtained when using $\Delta K_{eq}$ (see Figure 12). In these cases, the occurrence of some crack face contacts was also considered.

Murakami et al. [24] studied the fatigue crack behavior of S45C steel under pure mode II and mode III loading. Because the fractographic observations after each experiment revealed strong similarities, they concluded that the mechanisms of mode II and III shear FCG are essentially the same. When comparing the fracture surfaces obtained in this study under mixed mode I/III loading with those from mixed mode I/II loading [25], it is clear that they were very similar. In particular, no clear striation patterns were found near the crack tip region in both the cases. Moreover, the coplanar cracks branched when $\delta$ was increased to 120° and the loading ratio was 1.5, regardless of $\Delta K_{II}/\Delta K_I$ or $\Delta K_{III}/\Delta K_I$, for both loading cases. WT exhibited the fastest coplanar FCG rates and, among the rail steels, the rates for RP were higher than those for RF when plotted against their appropriate equivalent SIF ranges, for both loading cases. Furthermore, Akama and Kiuchi [11] reported that FCG rates in RP under these two loading conditions (mixed modes I/II and I/III) were almost equal when plotted against the SIF ranges considered to be the main driving forces, $\Delta K_{II}$ and $\Delta K_{III}$ for the mixed mode I/II and I/III loading conditions, respectively. Therefore, the mechanism of shear-mode FCG under non-proportional mixed mode loadings that were subject to the RCF cracks can be considered to be the same even if the main crack driving force is in-plane shear or out-of-plane shear.

## 5. Conclusions

To determine the coplanar and branch FCG rates of normal rail, head hardened rail, and wheel steel, fatigue tests were conducted under non-proportional mixed mode I/III loading cycles that simulated the RCF conditions. SEM observations and FEA were also performed to investigate the FCG behavior. The results can be summarized as follows.

1.  In RP and WT, a coplanar growth was obtained when $\delta$ was smaller than or equal to 90°. The growth rates were relatively well correlated when plotted against $\Delta K_s$ defined by Equation (6). The highest coplanar FCG rate was observed in WT, followed by RP and then RF.

2. In the RP case, the branch FCG occurred at $\delta = 120°$. The growth rates were plotted against $\Delta K_{eq}$ defined by Equation (10), which considers the $\Delta K_{III}$ attenuation due to the crack face contact, giving a good correlation.

3. Based on the fracture surface observations by SEM and the FEA results, the growth of long coplanar cracks was assumed to be driven mainly by mode III loading.

4. The FEA results showed that RF, which is a high-tensile steel, had smaller CTODs during the loading cycles compared with RP. Therefore, contact was likely to occur between the crack faces owing to the surface irregularities, causing crack branching in RF even under the same conditions.

5. When $\delta$ increased, $\Delta\sigma_{max}$ with respect to $\Delta\tau_{max}$ also increased, and the $\Delta\sigma_{max}$ plane turned toward the branch direction. Therefore, it can be concluded that the cracks tend to branch when $\delta$ increases.

6. The comparison of the fracture surfaces, branching conditions, and coplanar FCG rates data under mixed mode I/III loading and those under mixed mode I/II loading [25] indicated that the coplanar crack growth mechanisms in these two loading cases were similar regardless of whether the main driving force was in-plane or out-of-plane shear.

**Author Contributions:** Conceptualization, M.A.; Methodology, M.A. and A.K.; Software, M.A.; Validation, M.A. and A.K.; Formal Analysis, M.A.; Investigation, M.A. and A.K.; Writing-Original Draft Preparation, M.A.; Writing-Review & Editing, M.A.; Supervision, M.A.

**Funding:** This research was performed as part of the research and development programme for the future of railways, entitled 'Creation of rail damage/ballast track deterioration models and evaluation of maintenance work saving technologies', and was funded by the Railway Technical Research Institute.

**Conflicts of Interest:** The founding sponsors had no role in the design of the study; in the collection, analyses, or interpretation of data; in the writing of the manuscript, and in the decision to publish the results.

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
