# Peer review of "Fatigue Crack Growth under Non-Proportional Mixed Mode Loading in Rail and Wheel Steel Part 2: Sequential Mode I and Mode III Loading"

_applsci, doi:10.3390/app9142866_

Round 1
Reviewer 1 Report
The paper is mostly well-written. However, proofreading the manuscript to eliminate spelling and grammar mistakes is recommended. The following concerns should be addressed before accepting this paper for publication
1) The specimen is not pre-cracked. None of the experimental evidence shows that the crack is advancing. The displacement captured by the clip gauges could be due to slip or noise. Can the authors provide a concrete evidence that the crack has actually formed and advanced? (stop-cut-see experiments/ x-ray tomography showing the crack growth etc)
2) The actual problem is with the usage of fracture mechanics techniques for a plastic material. There will be very large plastic strains at the crack tip and even beyond the crack tip depending on the stress amplitudes which were not mentioned in none of the diagrams. It is important that authors should provide more finite element analysis results with plastic strain contours to substantiate the use of elasto-plastic fracture mechanics techniques
3) The equations provided in section 2.5.1 need more context although they might be from old work and a clear explanation about why they were used should be provided.
4) What is the experimental technique used by authors to evaluate the crack direction? As of now, there are no universally accepted rules to determine the crack growth just by looking at the fracture surfaces. So, more substantiation or experimental results are necessary here.
5) The SEM images unequivocally show ductile and brittle cleavage fractures. There are several fracture models for these types of fracture which can accurately track the growth of cracks. What is the necessity to use the approach adopted in this study?
6) The stress strain curves used for finite element analysis are missing and necessary for re-generating the results
7) What do authors mean by “Magnification” in fig 17 caption? Deformation amplification?
8) How can we infer from SEM images that the coplanar cracks are driven by mode 111 loading (conclusion-3)?
Author Response
The paper is mostly well-written. However, proofreading the manuscript to eliminate spelling and grammar mistakes is recommended. The following concerns should be addressed before accepting this paper for publication.
Response : Our paper has been verified by Enago English Proofreading Service. If you found spelling and grammar mistakes, please indicate “where”, “how”, and “why”.
Point 1: The specimen is not pre-cracked. None of the experimental evidence shows that the crack is advancing. The displacement captured by the clip gauges could be due to slip or noise. Can the authors provide a concrete evidence that the crack has actually formed and advanced? (stop-cut-see experiments/ x-ray tomography showing the crack growth etc)
Response 1: I will add the sentences as follows.
“Prior to the experiments, specimens containing notch depths of 5, 6, 7, and 8.5 mm were prepared, and the relation between load and output of the displacements from the gauges when each specimen was loaded up to 30 kN was investigated. In each case, a linear relation was obtained with no deviation, which was considered to be caused by slip. Additionally, the accuracy was verified using the potential drop method.”
Point 2: The actual problem is with the usage of fracture mechanics techniques for a plastic material. There will be very large plastic strains at the crack tip and even beyond the crack tip depending on the stress amplitudes which were not mentioned in none of the diagrams. It is important that authors should provide more finite element analysis results with plastic strain contours to substantiate the use of elasto-plastic fracture mechanics techniques.
Response 2: This study mainly deals with coplanar crack growth and the basic premise here is that the size of the plastic zone ahead of the crack front is small enough compared to the crack length, small scale yielding (SSY) condition, so that the stress intensities can be used. Actually, as the reviewer pointed out, this SSY condition may be violated around the latter coplanar crack growth data in each experiments, because a wide range of crack depths was covered. Wong et al. [18] performed the experiments to obtain the fatigue crack growth rates data under sequential mixed-mode I and II loading cycles and the experiments sometimes beyond the SSY condition. If we consider the deviation from the SSY condition in some experimental results, we cannot unify and organize the results by the stress intensity factor. The equivalent strain distribution near the crack front at the maximum tension in a loading cycle is shown as an example for the case of ΔKIII/ΔKI = 1.5, δ = 90 degree. In this case, the crack length (depth) is 4.6 mm and the SSY is estimated about less than 0.1 mm, which sufficiently satisfies the SSY condition.
Point 3: The equations provided in section 2.5.1 need more context although they might be from old work and a clear explanation about why they were used should be provided.
Response 3: Certainly these equations have been proposed a while ago, but are sufficiently reliable and are used frequently until now.
Point 4: What is the experimental technique used by authors to evaluate the crack direction? As of now, there are no universally accepted rules to determine the crack growth just by looking at the fracture surfaces. So, more substantiation or experimental results are necessary here.
Response 4: The correct crack growth direction is not measured. However, whether it is coplanar or branch growth can be clearly confirmed just looking at the fracture surface.
Point 5: The SEM images unequivocally show ductile and brittle cleavage fractures. There are several fracture models for these types of fracture which can accurately track the growth of cracks. What is the necessity to use the approach adopted in this study?
Response 5: The purpose of the SEM observation is to investigate whether the crack mainly grows by mode I loading or mode III loading. Because the main features of the fracture surface are rubbed-out ridges and valleys, this suggests that a mode I driving mechanism is questionable.
Point 6: The stress strain curves used for finite element analysis are missing and necessary for re-generating the results.
Response 6: I am sorry I forgot to add the Table. I will add the Table 4.
Point 7: What do authors mean by “Magnification” in fig 17 caption? Deformation amplification?
Response 7: That is right, thank you. I will change the word ” “Magnification” to “Deformation amplification”.
Point 8: How can we infer from SEM images that the coplanar cracks are driven by mode 111 loading (conclusion-3)?
Response 8: One common feature observed on fracture surfaces under mode I loading was striation formation due to the crack tip blunting and re-sharpening, where the spacing between the striation marks represented the rate of crack tip extension. However, as we mentioned in 2.5.3, no clear striation patterns were found near the crack tip region. If the main driving force is a mode I mechanism, traces of striation marks near the crack tip should be observed. Because the main features of the fracture surface are rubbed-out ridges and valleys, this suggests that a mode I driving mechanism is questionable. There is insufficient evidence to conclude that the crack growth is not related to striation mechanism, but observations of fracture surface sliding or rubbing suggests that the damage near the crack tip is likely to be caused by the mode III loadings.
In addition to the questions from the reviewers, there are some parts that replaced the reviews of the paper.
FEA was re-calculated by general purpose code MARC.
Some sentences were modified by Enago English Proofreading Service.

Reviewer 2 Report
The paper is related with the fatigue crack growth under non-proportional mixed mode loading in rail and wheel steel, including finite element analysis, as well as fractographic analysis of the failure surfaces.
The paper is quite well understandable. Appears some detailed and well explained information about previous research.
Correctly describes the state of the problem and associated issues.
The paper is a good contribution to the knowledge transfer between research institutions and society.
This is an interesting paper and worth of publishing after minor amendments.
In my opinion the following changes are necessary:
- Why this kind of geometry (Figure 1)?
- Page 4, Line 122: Tables 1 and 2.
- Why this kind of loading history (mode I and mode III)?
- The authors need to add some figures from the FEA (section 3).
Author Response
Point 1: Why this kind of geometry (Figure 1)?
Response 1: Circumferentially notched round bar specimen is the most common shape for crack growth testing involving Mode III loading.
Point 2: - Page 4, Line 122: Tables 1 and 2.
Response 2: ???
Point 3: Why this kind of loading history (mode I and mode III)?
Response 3: This loading history simulated the one experienced by rolling contact fatigue cracks in the presence of a fluid, as obtained by FEA [1,3].
Point 4: - The authors need to add some figures from the FEA (section 3).
Response 4: We will add the new section “3.2.2. Planes of maximum principal stress” and the related figures and tables are also added.
In addition to the questions from the reviewers, there are some parts that replaced the reviews of the paper. FEA was re-calculated by general purpose code MARC.
Some sentences were modified by Enago English Proofreading Service. Enago English Proofreading Service.

Reviewer 3 Report
Dear Authors, I found your work very interesting as it deals with mixed mode and experimental work to support the understanding of the mechanisms involved. I believe the paper is well written and organised, and in order to make it even better I would like you to consider the following comments:
1. Lines 92-98 clarify whether the description refers to the capability of the load cell and the machine or the testing conditions.
2. Lines from 99 explain whether the rotation of the two ends of the specimen under the torsional load affects the opening. Moreover, the testing conditions should be clarified as it are not clear.
3. Figure 1, is the specimen cylindrical or square?
4. From line 143 onward use references for the equations
5. Lines 177 onward it could be useful to highlight the features discussed on the fracture surfaces.
6. Equation 203 should be explained with more details as it is not clear which crack growth you are referring to. Is it the total? Moreover, which load ratio is it referring to?
7. You are fitting the FCG curves for each material with a single straight line although the fitting is poor. The R2 is high because the two group of data are symmetric but they should be fitted with different curves.
8. Can you provide the justification and the sources for the equation used to work out the mixed mode SIF range?
9. Can you compare the prediction from the FEA with the Experiments to quantify the error in the modelling approach?
10. Are you modelling the crack propagation process? Is it an iterative model? Are you using the Strain approach?
11. I would suggest to show fig 17 as contour plot of total displacement.
12. Lines 436-451 not clear the point that you are making and the use of the results from [23].
13. Some of the conclusions (4, 5 and partially 1) do not seem to be linked to the discussion. Can you add some details in the main structure to make it clear.
Author Response
Point 1: Lines 92-98 clarify whether the description refers to the capability of the load cell and the machine or the testing conditions.
Response 1: The description refers to the capability of the machine. I will change and add the sentences to clarify the facts.
“The fatigue tests were performed on a servo-hydraulic testing machine, in which capabilities are a maximum tension–compression force of 200 kN, a maximum fully reversed torque of 1,000 Nm under load control, and a frequency of 1 Hz in dry conditions.”
Point 2: Lines from 99 explain whether the rotation of the two ends of the specimen under the torsional load affects the opening. Moreover, the testing conditions should be clarified as it are not clear.
Response 2: I will add the sentences as follows.
“Prior to the experiments, specimens containing notch depths of 5, 6, 7, and 8.5 mm were prepared, and the relation between load and output of the displacements from the gauges when each specimen was loaded up to 30 kN was investigated. In each case, a linear relation was obtained with no deviation, which was considered to be caused by slip. Additionally, the accuracy was verified using the potential drop method.”
Point 3: Figure 1, is the specimen cylindrical or square?
Response 3: The specimen is cylinder. Line 94 and the caption of Fig.1 says “round bar specimen” .
Point 4: From line 143 onward use references for the equations
Response 4: I am sorry if I misunderstood your meaning, but equation (1), (2), (3) and (4) are from reference [13], (5) and (6) are from reference [14], (9) from [15], (10) from [16], all equations referred literatures. If I misunderstand your meaning, please indicate again.
Point 5: Lines 177 onward it could be useful to highlight the features discussed on the fracture surfaces.
Response 5: We already discussed the features of fracture surfaces from line 180 to 186. We think that these are sufficient for the macroscopic fractures as shown in Fig.6.
Point 6: Equation 203 should be explained with more details as it is not clear which crack growth you are referring to. Is it the total? Moreover, which load ratio is it referring to?
Response 6: Equation (7) was applied to all these data RP including RP1 for ΔKIII/ΔKI = 1.0,δ = 30, 90, RP2 for ΔKIII/ΔKI = 1.5,δ = 90, and RP3 for ΔKIII/ΔKI = 1.5,δ = 30. I will add “ to all these data” to the sentence .
Point 7: You are fitting the FCG curves for each material with a single straight line although the fitting is poor. The R2 is high because the two group of data are symmetric but they should be fitted with different curves.
Response 7: We thank reviewer to point out the important fact concerning the R2. The two group of data are fitted with different curves and the sentences are modified.
Point 8: Can you provide the justification and the sources for the equation used to work out the mixed mode SIF range?
Response 8: As I answered for Question 4, equation (1), (2), (3) and (4) are from reference [13], (5) and (6) are from reference [14], (9) from [15], (10) from [16], all equations referred literatures.
Point 9: Can you compare the prediction from the FEA with the Experiments to quantify the error in the modelling approach?
Response 9: As we mentioned in section 3.1, this mesh and the loading procedure confirmed that the nominal stress value in the ligament ahead of the crack was accurate.
Point 10: Are you modelling the crack propagation process? Is it an iterative model? Are you using the Strain approach?
Response 10: No. We considered a stationary crack because, in this study, a branch angle from the main crack was predicted. I will add the sentences in Section 3.1.
“A series of elasto-plastic FEA was performed using the commercial FEA code MARC to elucidate the FCG characteristics observed in the mixed mode I/III loading tests and predict the FCG direction, which is θ and φ in Fig.5. Therefore, a stationary crack was considered.”
Point 11: I would suggest to show fig 17 as contour plot of total displacement.
Response 11: Thank you for your advice. Fig.17 will be changed to the contour plot of total displacement.
Point 12: Lines 436-451 not clear the point that you are making and the use of the results from [23].
Response 12: The following sentence will be added to clarify the point.
“In particular, no clear striation patterns were found near the crack tip region in both cases.”
Point 13: Lines 436-451 not clear the point that you are making and the use of the results from [23].
Response 13: Conclusion (4) is based on the following sentences in the Discussion.
“The FEA results showed that the CTOD variation in one loading cycle was smaller for RF than for RP under the same loading condition. In non-proportional mixed mode loading cycles, fatigue cracks follow the direction of the Δσmax or Δτmax plane depending on these growth rates on these planes is faster [23]. When the CTOD is small, the actual crack faces have irregularities and, hence, are likely to contact. When the crack faces contact, the friction can make the coplanar crack growth rate smaller relative to the branch crack growth rate. Therefore, factory-roof fractures, which are traces of branch crack growth, should have appeared.”
And conclusion (5) is based on the following sentences in the Discussion.
“When δ increased to 120°, branch cracks also grew even in RP. According to the FEA results shown in Fig. 19, increasing δ increased Δσmax with respect to Δτmax and turned it toward the branch direction. On the other hand, Δτmax remained on the coplanar direction. The coplanar FCG rates under the loading cycles adopted in this study are considered to have a strong dependence on KIII. Therefore, it should also have a relatively strong dependence on Δτmax, while the branch FCG rates are generally considered to have a relation with Δσmax. From these facts, it can be concluded that cracks tend to branch when δ increases.”
In addition to the questions from the reviewers, there are some parts that replaced the reviews of the paper.
FEA was re-calculated by general purpose code MARC.
Some sentences were modified by Enago English Proofreading Service.

Reviewer 4 Report
The article presents numerical and experimental investigations on notched round bar specimens undergoing partially superimposed Mode I and Mode III loading. The article is interesting and the topic is appropriate for publishing in this journal. However, there are several aspects in which the manuscript has to be improved before accepting it.
Paragraph 3 must be improved. Some more tables/pictures/plots must be added to let the reader better understand what was explicitely modelled numerically, how the numerical campaign was conducted, how the loads were applied: do they are cyclic? how many cycles applied? etc.
Paragrapgh 4 is almost incomprehensible. It is not clear which results have been obtained. Perhaps this is also due to the difficulties in the understanding of paragrapgh 3.
Also the conclusions should be more brief and concise.
Some further suggestions in the followings:
- lines 20-24: the sentence should be rephrased;
- line 42: please see the following reference to improve this aspect:
V. Giannella, G. Dhondt, C. Kontermann, R. Citarella, Combined static-cyclic multi-axial crack propagation in cruciform specimens, International Journal of Fatigue, 123 (2019), 296-307.
- Table 4 with material properties is missing.
Author Response
Point 1: Paragraph 3 must be improved. Some more tables/pictures/plots must be added to let the reader better understand what was explicitely modelled numerically, how the numerical campaign was conducted, how the loads were applied: do they are cyclic? how many cycles applied? etc.
Response 1: Thank you for your advice. In line with your advice and other reviewer’s advices, some sentences will be added in section 3 as follows.
“A series of elasto-plastic FEA was performed using the commercial FEA code MARC to elucidate the FCG characteristics observed in the mixed mode I/III loading tests and predict the FCG direction, which is θ and φ in Fig.5. Therefore, a stationary crack was considered.”
“At the center of the upper surface of the mesh, a pilot node was connected to the remaining nodes of that surface through a multipoint constraint; force (P) and torque (Ta) were applied to the node”
“The contour plots of total displacement on the shapes deformed under the positions of the loading cycle for the RP case are shown in Fig. 17.”
The new section “3.2.2. Planes of maximum principal stress” is added and the related figures and tables are also added. Also the sentences concerning the section will be added in “4. Discussion”.
Point 2: Paragrapgh 4 is almost incomprehensible. It is not clear which results have been obtained. Perhaps this is also due to the difficulties in the understanding of paragrapgh 3.
Response 2: Along the advices by all reviewers, section 4 is modified. They are indicated by coloured letters. For example,
“In the case of RP and WT, a coplanar growth was obtained when δ was smaller than or equal to 90º regardless of the ΔKIII /ΔKI value and the growth rate was well correlated with ΔKs as shown in Fig.9.”
“The FEA results shown in Fig.21 indicated that the CTOD variation in one loading cycle was smaller for RF than for RP under the same loading condition.”
…
…
We hope that the section became easier to understand. If there are same parts that are still unclear, please point out “where”, “what” and “how” again.
Point 3: Also the conclusions should be more brief and concise.
Response 3: Because many readers mainly read the conclusions, please approve summarizing conclusions to these extents.
Point 4: lines 20-24: the sentence should be rephrased;
Response 4: The sentence will be changed as follows.
“Based on the results obtained in this study, the mechanism of long shear-mode crack growth turned out to be the same regardless of whether the main driving force is in-plane shear or out-of-plane shear.”
Point 5: please see the following reference to improve this aspect:
V. Giannella, G. Dhondt, C. Kontermann, R. Citarella, Combined static-cyclic multi-axial crack propagation in cruciform specimens, International Journal of Fatigue, 123 (2019), 296-307.
Response 5: We will introduce the recommended paper in “1. Introduction” as follows.
“Giannella et al. [9] investigated the FCG behavior in cruciform specimens made of Ti6246 by applying a static load along one arm of the specimen and a cyclic load along the other arm. They considered all mode I/II/III loading and determined that a change in the FCG direction occurred depending on the static load levels.”
Point 6: Table 4 with material properties is missing.
Response 6: I am sorry I forgot to add the Table. I will add the Table.4.
In addition to the questions from the reviewers, there are some parts that replaced the reviews of the paper.
FEA was re-calculated by general purpose code MARC.
Some sentences were modified by Enago English Proofreading Service.

Round 2
Reviewer 1 Report
Accept as it is
Author Response
I am very pleased to be reviewed by the expert in this research field. I felt conscience in your tough review.
Reviewer 4 Report
The article has been improved but some more modifications should be performed before publishing, e.g.:
line 60: cruciform specimens are loaded in-plane, therefore it seems unlikely that they achieved non-negligible mode III. Should it be "I/II modes" instead of "all mode I/II/III" ?
line 322: which are the angles ...
figure 17c: it should be better to show a stress field (e.g. von Mises?) instead of showing mostly the same already shown in fig. 17b.
line 503: typo as -> was.
In final, conclusions should be reduced in size, neglecting all the unnecessary details. Consider also to accept the following suggestions for conclusions:
(1): it could be reduced as: "In RP and WT, an in-plane growth was obtained when δ was smaller than or equal to 90°; growth rates were relatively well correlated when plotted against the ΔKeq defined by Eq. 10.".
(2): In the RP case, a branch FCG occurred for δ = 120°. Growth rates were plotted against ΔKt, but however a good correlation was obtained only when plotted against ΔKeq, since crack face contact was considered by the latter.
(3) it is really unlikely that mode III dominance leads to in-plane crack growths. E.g. see the "torque load case" in: R. Citarella, V. Giannella, M. Lepore, G. Dhondt, Dual boundary element method and finite element method for mixed-mode crack propagation simulations in a cracked hollow shaft, Fatig. Fract. Eng. Mater. Struct., 41 (2018), pp. 84-98.
(4) it seems unlikely that small crack face contacts leads to relevant kinking of the crack. Please double check.
(5) This was expectable and it could be omitted in the conclusions.
(6) Mode I/II results are not available in the text and should be at least referenced. However, it seems unlikely that cracks undergoing mode I/II conditions behave like if undergoing mode I/III.
Author Response
Point 1: Line 60: cruciform specimens are loaded in-plane, therefore it seems unlikely that they achieved non-negligible mode III. Should it be "I/II modes" instead of "all mode I/II/III" ?
Response 1: Since this paper is about mixed Mode I / III, it is not appropriate to review the paper about mixed Mode I / II. The paper by Giannella et al. applied Yaoming-Mi formula to obtain an equivalent K-value that can consider all mode I/II/III. Therefore, I want to rewrite the sentence as follows.
“Giannella et al. [9] investigated the FCG behavior in cruciform specimens made of Ti6246 by applying a static load along one arm of the specimen and a cyclic load along the other arm. They used an equivalent stress intensity range in the Walker crack growth law that can consider all mode I/II/III loading and determined that a change in the FCG direction occurred depending on the static load levels.”
If the reviewer is still unconvinced and willing to consider alternatives, please do not hesitate to inquire.
Point 2: line 322: which are the angles ...
Response 2: This “which” is singular because it receives a “direction”. This seems to be Enago's way of thinking. If you think this is a mistake or I misunderstand, please do not hesitate to inquire.
Point 3: figure 17c: it should be better to show a stress field (e.g. von Mises?) instead of showing mostly the same already shown in fig. 17b.
Response 3: Thank you for your good advice. I will change Fig.17(c) from total displacement to tz-stress, because the stress is closely related to mode III loading.
Point 4: line 503: typo as -> was.
Response 4: Thank you for finding our mistake. I will fix it. However, the sentence is modified.
Point 5: In final, conclusions should be reduced in size, neglecting all the unnecessary details. Consider also to accept the following suggestions for conclusions:
Point 5 (1): it could be reduced as: "In RP and WT, an in-plane growth was obtained when δ was smaller than or equal to 90°; growth rates were relatively well correlated when plotted against the ΔKeq defined by Eq. 10.".
Response 5 (1): Thank you very much. I will modify in line with your advice as follows.
“(1) In RP and WT, a coplanar growth was obtained when δ was smaller than or equal to 90°; the growth rates were relatively well correlated when plotted against the ΔKs defined by Eq. (6). The highest coplanar FCG rate was observed in WT, followed by RP and then RF.”
Point 5 (2): In the RP case, a branch FCG occurred for δ = 120°. Growth rates were plotted against ΔKt, but however a good correlation was obtained only when plotted against ΔKeq, since crack face contact was considered by the latter.
Response 5 (2): Thank you very much. I will modify in line with your advice as follows.
“(2) In the RP case, the branch FCG occurred at δ = 120°. The growth rates were plotted against ΔKeq defined by Eq.(10), which considers the ΔKIII attenuation due to the crack face contact, giving a good correlation.”
Point 5 (3): it is really unlikely that mode III dominance leads to in-plane crack growths. E.g. see the "torque load case" in: R. Citarella, V. Giannella, M. Lepore, G. Dhondt, Dual boundary element method and finite element method for mixed-mode crack propagation simulations in a cracked hollow shaft, Fatig. Fract. Eng. Mater. Struct., 41 (2018), pp. 84-98.
Response 5 (3): In this regard, I would like to refute the reviewer. Indeed, when the loading is pure Mode III, or cyclic Mode III with static Mode I, the cracks often branch. However, when the loading is sequential overlapped mixed Mode I and Mode III loading, which is shown in Fig.4, long coplanar crack growth occurs definitely. When the coplanar crack growth rates were plotted against ΔKI, the FCG rates in mixed Mode I and Mode III appear to be much quicker than the corresponding rates in pure Mode I. Please refer our paper “Long coplanar mode III fatigue crack growth under non-proportional mixed mode loading in rail steel. Proc. IMechE, Part F: J. Rail Rapid Transit, 2012, 226, 489–500”.
I will read the recommended paper in detail for my future research.
Point 5 (4): it seems unlikely that small crack face contacts leads to relevant kinking of the crack. Please double check.
Response 5 (4): Certainly, this is not verified enough and it is only the conjecture.
Point 5 (5): This was expectable and it could be omitted in the conclusions.
Response 5 (5): I think that “When δ increased, Δσmax with respect to Δτmax also increased and the Δσmax plane turned toward the branch direction” is not necessarily obvious for the sequential mixed mode I and mode III loading adopted in this study. However, “Because the branch FCG rates are generally considered related to the tangential stress range” is expectable. So the conclusion (5) will be modified as follows.
“When δ increased, Δσmax with respect to Δτmax also increased and the Δσmax plane turned toward the branch direction. Therefore, it can be concluded that the cracks tend to branch when δ increases.”
Point 5 (6): Mode I/II results are not available in the text and should be at least referenced. However, it seems unlikely that cracks undergoing mode I/II conditions behave like if undergoing mode I/III.
Response 5 (6): I will add the reference number.
As for the reviewer’s doubt, we are referring to the coplanar crack growth under the sequential overlapped mixed Mode I and Mode III loading adopted in this paper. Of course, this is not the case for pure Mode III and Mode II loading, or for cyclic Mode III loading with static Mode I loading. The term “coplanar crack” is added to emphasize that it is for the coplanar crack growth. Corresponding to this, the same term is added in the abstract.
I hope we can have enough discussions someday and somewhere (for example, International Conference on Multiaxial Fatigue).
